# Early Low-Level Arsenic Exposure Impacts Post-Synaptic Hippocampal Function in Juvenile Mice

**DOI:** 10.3390/toxics9090206

**Published:** 2021-08-31

**Authors:** Karl F. W. Foley, Daniel Barnett, Deborah A. Cory-Slechta, Houhui Xia

**Affiliations:** 1Department of Neuroscience, University of Rochester Medical Center, Rochester, NY 14642, USA; Houhui_Xia@URMC.Rochester.edu; 2Department of Pharmacology & Physiology, University of Rochester Medical Center, Rochester, NY 14642, USA; dmb4001@med.cornell.edu; 3Department of Environmental Medicine, University of Rochester Medical Center, Rochester, NY 14642, USA; Deborah_Cory-slechta@urmc.rochester.edu

**Keywords:** arsenic, synaptic transmission, long-term potentiation, hippocampus, development

## Abstract

Arsenic is a well-established carcinogen known to increase mortality, but its effects on the central nervous system are less well understood. Epidemiological studies suggest that early life exposure is associated with learning deficits and behavioral changes. Studies in arsenic-exposed rodents have begun to shed light on potential mechanistic underpinnings, including changes in synaptic transmission and plasticity. However, previous studies relied on extended exposure into adulthood, and little is known about the effect of arsenic exposure in early development. Here, we studied the effects of early developmental arsenic exposure in juvenile mice on synaptic transmission and plasticity in the hippocampus. C57BL/6J females were exposed to arsenic (0, 50 ppb, 36 ppm) via drinking water two weeks prior to mating, with continued exposure throughout gestation and parturition. Electrophysiological recordings were then performed on juvenile offspring prior to weaning. In this paradigm, the offspring are exposed to arsenic indirectly, via the mother. We found that high (36 ppm) and relatively low (50 ppb) arsenic exposure both decreased basal synaptic transmission. A compensatory increase in pre-synaptic vesicular release was only observed in the high-exposure group. These results suggest that indirect, ecologically relevant arsenic exposure in early development impacts hippocampal synaptic transmission and plasticity that could underlie learning deficits reported in epidemiological studies.

## 1. Introduction

Early life exposure to toxic chemicals and environmental pollutants is associated with learning deficits and behavioral changes [1,2,3]. An estimated 200 million people worldwide are exposed to arsenic concentrations in drinking water that exceed the World Health Organization’s recommended limit, 10 parts per billion (ppb) [4]. Exposure to concerning levels of arsenic is not limited to toxic waste sites. Rather, arsenic levels commonly exceed 10 ppb in domestic wells throughout the United States, especially in the southwest. While arsenic levels are kept below 10 ppb in municipal water supplies, private wells are unregulated and arsenic levels exceed 10 ppb in 20 out of 37 principal aquifers in the United States [5]. Even mild increases in arsenic exposure are of concern, as exposure is associated with numerous adverse health outcomes and increased mortality from a variety of conditions, including cardiovascular disease and cancer, as well as increased infant mortality [4]. Further, recent studies suggest the consequences of arsenic exposure can span across generations [6,7,8].

In January 2006, the maximum contaminant level (MCL) of arsenic in public water systems was lowered from 50 to 10 ppb, in compliance with a previous United States Environmental Protection Agency (EPA) ruling [9]. This change was enacted due to several epidemiological studies demonstrating an increased risk of cancer. While acute, high-level arsenic exposure was known to be associated with peripheral neuropathy, relatively little was known about the neurological consequences of chronic, low-level arsenic exposure [National Research Council (NRC) [10]]. However, more recent epidemiological studies have demonstrated that arsenic exposure is associated with deficits in cognitive and motor functions in children and adults [11,12,13,14,15,16,17,18]. Additionally, a recent study suggests inequalities in arsenic exposure reductions following the 2006 change in the MCL, such that there was a higher concentration of arsenic in public water systems serving Hispanic and tribal communities, small rural communities, and southwestern U.S. communities [19]. Given the relatively recent change in the arsenic MCL in public water in the U.S., the continued high exposure in many regions worldwide and the known vulnerability of the developing brain to toxicants and pollutants underscores the critical need to understand the effects of early life exposure to arsenic.

Electrophysiological studies in arsenic-exposed rodent models have begun to shed light on the potential mechanistic underpinnings of the associated cognitive deficits. Rodents exposed to high arsenic concentrations throughout early development and adulthood demonstrate a decrease in synaptic transmission and long-term potentiation (LTP) [20] in the hippocampus that may be secondary to altered glutamate transport [21]. The findings suggest that both pre- and post-synaptic functions are altered due to compensatory changes, since basal synaptic transmission decreased following arsenic exposure, whereas the neurotransmitter release probability increased. These changes are also reflected by changes in the expression of excitatory receptors and other regulators of synaptic signaling [22,23,24,25,26]. Similar changes in synaptic transmission and plasticity have been demonstrated by the ex vivo exposure of hippocampal slices to arsenite metabolites [27,28]. However, the electrophysiological effects of early developmental arsenic exposure have not been distinguished from chronic adulthood exposure. As arsenic can cross the placenta [29], we reasoned that early developmental arsenic exposure could cause changes in synaptic transmission and plasticity even without adulthood exposure. Specifically, given the epidemiological evidence of neurobehavioral deficits in children, we hypothesized that early developmental arsenic exposure alone reduces synaptic transmission and plasticity. Still, the effects of continued adulthood exposure on hippocampal synaptic function could differ from the effects of early developmental exposure alone. For example, whereas acute ex vivo exposure to arsenic metabolites attenuates LTP in hippocampal slices from adult rats, it facilitated LTP in young rats [30]. Further, there are pre- and post-synaptic changes present in adult mice following chronic exposure that suggest compensatory changes in the hippocampal circuit, with a decrease in basal synaptic transmission but an increase in neurotransmitter release probability [20]. However, the directionality of these synaptic changes is unclear, e.g., does a decrease in post-synaptic receptors cause a compensatory increase in pre-synaptic neurotransmitter release, or vice versa? Here, we studied the effects of in vivo arsenic exposure during gestation and early development by exposing dams to a high level (36 ppm) or a low level of arsenic (50 ppb), the MCL for public drinking water in the United States prior to 2006. Of note, in this paradigm, the dam is exposed to arsenic directly via drinking water, whereas the offspring are exposed indirectly via maternal transmission. Surprisingly, we found that the maternal exposure to even low levels of arsenic, i.e., 50 ppb, impairs synaptic transmission in the hippocampus of the offspring. Additionally, we observed different effects of arsenic exposure in our juvenile mice than what has been previously reported in adult mice (Table 1 and Table 2).

## 2. Materials and Methods

### 2.1. Arsenic Exposure

All experimental protocols were approved by the Institutional Animal Care and Use Committee of the University of Rochester and carried out in compliance with ARRIVE guidelines. Given that arsenic(V) acid salt (arsenate) is the most common form of arsenic in groundwater (Cullen and Reimer, 1989), we utilized sodium arsenate dibasic heptahydrate (Na2HAsO4 7H2O; hereon, arsenic), obtained from MilliporeSigma (A6756). C57BL/6J females were exposed to arsenic (0, 50 ppb, 36 ppm) in their drinking water (distilled deionized H2O) starting at six weeks of age. Breeding began at two months of age and arsenic exposure continued after parturition to simulate protracted human exposure conditions. The juvenile offspring were then used for experiments prior to weaning (P17–P23), such that the pups are still nursing-dependent. Both male and female pups were used for experiments. At least two mice from two litters per exposure group were used for each experiment. Mice were maintained with a 12:12 h light:dark cycle, constant temperature of 23 °C and ad libitum feeding. An overview of arsenic exposure is shown in the Graphical Abstract. Fresh arsenic solutions were prepared and exchanged every 2–3 days to avoid oxidation.

### 2.2. Electrophysiology

Acute hippocampal slices were prepared from male and female juvenile mice (P17–P23) prior to weaning. After decapitation and rapid extraction of the brains into ice-cold artificial cerebrospinal fluid (ACSF), 400-micrometer-thick hippocampal slices were prepared. Slices were then allowed to recover in room temperature (RT) ACSF for at least one hour prior to experiments. Field recordings were conducted at Schaffer collateral-CA1 synapses in RT ACSF at a flow rate of 2–3 mL/min. A borosilicate recording electrode (1–3 MΩ) filled with 1 M NaCl was placed in CA1 stratum radiatum and a monopolar stimulating electrode was placed on Schaffer collaterals between CA3 and CA1. The ACSF solution consisted of, in mM: 120.0 NaCl; 2.5 KCl; 2.5 CaCl2; 1.3 MgSO4; 1.0 NaH2PO4; 26.0 NaHCO3; and 11.0 D-glucose. ACSF was aerated with carbogen (95% O2, 5% CO2) throughout slice preparation, incubation, and recordings. Responses were elicited every 15 s. Basal synaptic transmission was assessed using input–output (IO) curves, comparing the fiber volley, a more direct measure of axonal stimulation, to the field excitatory post-synaptic potential (fEPSP) slope. Short-term pre-synaptic plasticity was assayed using paired-pulse facilitation (PPF) of varying inter-pulse intervals. Finally, LTP was induced by a single tetanus of one second, 100 Hz stimulation. The following sample sizes were used for each experiment (Group: mice, slices): IO (Control: 5, 10; 36 ppm: 6, 13; 50 ppb: 7, 15); PPF (Control: 5, 12; 36 ppm: 6, 14; 50 ppb: 8, 18); LTP (Control: 4, 7; 36 ppm: 6, 13; 50 ppb: 6, 7). Electrophysiology recordings were collected with a MultiClamp 700A amplifier (Axon Instruments, San Jose, CA, USA), PCI-6221 data acquisition device (National Instruments, Austin, TX, USA), and Igor Pro 7 (Wavemetrics, Portland, OR, USA) with a customized software package (Recording Artist, http://github.com/rgerkin/recording-artist, last commit 31 October 2019).

### 2.3. Analysis

Electrophysiology data was analyzed using R (version 4.0.2) and GraphPad Prism (version 9.2.0). To generate IO curves, fiber volley amplitudes were binned at ±0.05 mV, with the exception of 0.025 (0,0.025), 0.05 (0.025,0.05), and 0.1 mV (0.075,1.5). To measure the magnitude of LTP, we normalized the last five minutes of fEPSPs (25–30 min after LTP induction) to the 10 min baseline. Males and females were pooled for all analyses, with no sub-analysis of the effect of sex. Statistical significance between means was calculated using *t*-tests or two-way ANOVAs and Dunnett post hoc comparisons, with arsenic exposure and stimulation (fiber volley; inter-pulse interval) as factors.

## 3. Results

Juvenile mice (P17–P23) exposed to either a low (50 ppb) or high level (36 ppm) of arsenic in utero exhibit a significant decrease in basal synaptic transmission in the hippocampus at the Schaffer collateral-CA1 synapse (Figure 1A; two-way ANOVA, arsenic exposure: F(2,216) = 22.31, *p* < 0.0001; fiber volley: F(7,216) = 99.87, *p* < 0.0001). Interestingly, low and high arsenic exposure levels result in a similar decrease, such that high exposure does not reduce transmission beyond the deficit seen with low-level exposure. However, the two arsenic exposure levels differ in their effects on short-term pre-synaptic plasticity, as assessed by paired-pulse facilitation (PPF) (Figure 1D). High arsenic exposure reduced the PPF at short inter-pulse intervals (both 25 ms and 15 ms IPI, two-way ANOVA, arsenic exposure: F(2,242) = 11.93, *p* < 0.0001; IPI: F(5,242) = 49.22, *p* < 0.0001; Dunnett’s post hoc, *p* < 0.05), whereas there is no significant change from the control with low-level arsenic exposure. There was no significant difference between the groups at longer inter-pulse intervals.

To assess the long-term changes in synaptic plasticity, we gave high-frequency stimulation to induce LTP, a cellular model for neural circuit development as well as learning and memory. Surprisingly, high arsenic exposure levels increased LTP by about 11% (Figure 2; 36 ppm: 29% ± 0.03, *n* = 13; control: 18% ± 0.04, *n* = 7, *p* < 0.05). Developmental exposure to low-level arsenic led to an 8% increase in LTP compared to the control (50 ppb: 26% ± 0.06, *n* = 7), falling between the control and high arsenic exposure, but did not reach statistical significance.

## 4. Discussion

Our findings show that early developmental arsenic exposure results in significant changes to hippocampal synaptic transmission and plasticity. Most strikingly, even a relatively low dose of arsenic exposure decreases synaptic transmission. This is accompanied by no change in PPF, an indicator of glutamate release in presynaptic neurons; in our case, CA3 neurons. Therefore, our studies suggest that a low dose of arsenic led to a postsynaptic change in the CA1 neurons contributing to the decrease in synaptic transmission. This is consistent with the decrease in the neurite number and complexity observed in a cell culture model of arsenic exposure [31] as well as alterations in the hippocampal synaptic structure in arsenic-exposed juvenile mice [32]. On the other hand, a high level of arsenic decreased PPF, suggesting that glutamate release is increased in presynaptic CA3 neurons. However, there is no overall synaptic transmission change between two doses of arsenic (50 and 36 ppm). This is consistent with the notion that the higher concentration of arsenic caused a progression of changes to the hippocampal circuitry, such that there was a compensatory increase in glutamate release from presynaptic CA3 neurons in response to the changes in postsynaptic CA1 neurons for synaptic transmission.

Our data also indicate that both low and high levels of arsenic led to a trend of increased LTP expression, with an 8 and 11% increase, respectively. Our studies build upon and extend previous studies that utilize acute in vitro exposure and chronic in vivo adulthood exposure. First, our findings replicate the decrease in the hippocampal basal synaptic transmission observed both in acute, high-concentration in vitro exposure of arsenite metabolites in young rat hippocampal slices [27,28,30] as well as chronic, high-concentration (20 ppm) in vivo exposure in adult mice [20]. Second, we found that the decrease in PPF previously observed with a high-concentration arsenic exposure in adult mice [20] was also observed in our high-concentration exposure in juvenile mice (Figure 1D). Importantly, in our model, changes in basal synaptic transmission were observed even with low-concentration (50 ppb) gestational exposure. Third, our findings reveal differences between juvenile and adult mice exposed to arsenic. Whereas a previous study found a decrease in the degree of LTP in mice exposed to high-concentration arsenic from gestation through adulthood [20], we observed an increase in LTP in our juvenile mice. Together, these findings suggest a progression of changes induced by arsenic exposure, where LTP is facilitated in juvenile mice but attenuated with prolonged exposure, consistent with differential effects depending upon the timing of exposure, i.e., the critical window of exposure.

Given the epidemiological evidence [11,12,13,14,15,16,17,18], and the memory deficits observed in rodent models [11,12,20,33], we expected to see a decrease in LTP in the arsenic-exposure groups. The observed increase in LTP is difficult to reconcile, since the facilitation of LTP generally correlates with improvements in learning and memory in adults [34]. However, enhanced LTP has been observed in other rodent models of neurodevelopmental conditions, such as prenatal exposure to valproic acid, an insult-based animal model of autism [35]. In addition to an increase in LTP, valproic acid-exposed rodents show abnormal fear conditioning, decreased social interactions, and deficits in sensorimotor gating, which reflects an impairment in information processing or attention [36,37]. Therefore, while an increase in LTP represents an enhancement of learning and memory machinery at the cellular level, there may still be deficits at the behavioral level, especially when attention and information gating are impaired.

It is also important for the LTP data to be interpreted in the context of the changes in basal synaptic transmission and short-term pre-synaptic plasticity. Basal transmission is highly regulated, and alterations can lead to hippocampal circuit dysfunction. The decrease in basal transmission for both arsenic exposure groups could, therefore, lead to neurobehavioral impairments. Given that arsenic decreases neurite outgrowth in a neuronal cell culture model [31], the decrease in basal synaptic transmission in our juvenile mice may represent a decrease in hippocampus connectivity, i.e., a decrease in the number of functional excitatory dendritic spines. With fewer functional synapses, it is possible that LTP is preserved, whereas the capacity for the longer-term storage of information is greatly diminished: while short-term memory and LTP primarily depend on strengthening pre-existing synaptic connections, long-term storage is believed to require structural reorganization including the formation of new synapses [38,39,40,41,42]. A reduction in basal connectivity can, therefore, reduce the dynamic range of the structural reorganization necessary for the long-term storage of information. Relevant to this possibility, in two-month-old rats exposed to 3 ppm arsenic during gestation and after weaning, an impairment in contextual fear conditioning was only observed at 72 h after conditioning, but not at 1, 6, or 24 h [43].

Most strikingly, the pups in our experiments had little direct exposure to arsenic, rather they were exposed via the mother. Previous research suggests that the transmission of arsenic through the placenta is higher than the transmission into breast milk [29,44]. Therefore, it is likely that our results are primarily due to the gestational exposure in our paradigm. It is important to highlight the possible differences between the direct consumption of inorganic arsenic and gestational exposure. Inorganic arsenic undergoes biomethylation within the body, generating mono- and di-methylated organic arsenic metabolites. The metabolites possess different properties, toxicities, as well as rates of excretion [45,46]. Overall, methylated arsenicals are historically believed to be less acutely toxic than inorganic arsenic, though still biochemically active. The arsenic species that constitute gestational exposure will include both inorganic and methylated arsenicals, but primarily the methylated arsenicals according to studies of maternal and fetal cord blood [29,47]. While there is a strong correlation between the total arsenic concentration and the individual metabolite concentration in maternal and cord blood, one study in mice found that the accumulation of arsenicals in the newborn mouse brain was significantly higher than that of the exposed mother [48]. Therefore, although gestational exposure may decrease inorganic arsenic exposure compared to adult exposure, it is possible that arsenicals are more likely to pass the blood–brain barrier in the fetus.

One caveat of animal models is that biomethylation and excretion kinetics differ between species, even between mice and rats [49,50]. Indeed, even within species, it is possible that the age-related effects of arsenic could in part be due to differences in metabolism and excretion. Interestingly, among the different properties arsenic metabolites can possess, they can also have opposing effects on AMPA receptors and NMDA receptors, the major excitatory receptors underlying synaptic transmission and plasticity, respectively [30,51]. Relevant to this study, acute, ex vivo exposure to 1μM monomethylarsonous acid (MMA(III)) is the only metabolite and concentration that has been demonstrated to significantly increase LTP in young rats [30].

Overall, we found that early developmental arsenic exposure alters hippocampal synaptic transmission and plasticity in juvenile mice. Our findings generate intriguing questions regarding the age-related effects of gestational and chronic arsenic exposure, and how they might be altered. The 50 ppb findings are of particular interest for the following several reasons: (1) 50 ppb was the effective arsenic MCL in the United States for decades prior to 2006; (2) the permissible level of arsenic continues to be above 10 ppb in many countries; and (3) mean arsenic levels exceed 10 ppb in many common beverages and 50 ppb in many common foods [52]. The current results suggest that indirect, ecologically relevant arsenic exposure in early development impacts hippocampal synaptic transmission.

## Figures and Tables

**Figure 1 toxics-09-00206-f001:**
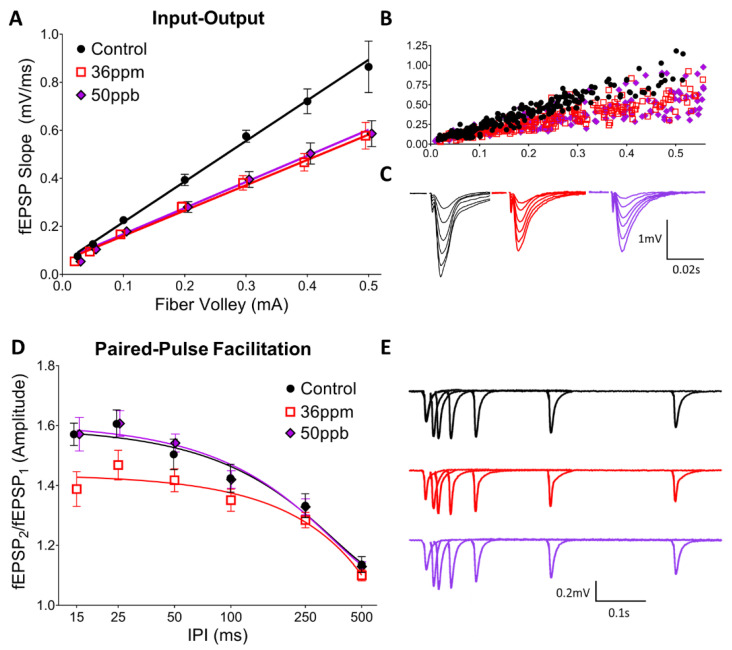
Effect of arsenic on hippocampal basal synaptic transmission (**A**–**C**) and short-term presynaptic plasticity (**D**,**E**), as assessed using input–output curves and paired-pulse facilitation, respectively. (**A**) Averaged responses for input-output experiments, (**B**) pre-binned, raw values from all experiments, and (**C**) waveforms averaged by fiber volley bin from representative experiments. (**D**) Averaged responses for paired-pulse facilitation experiments and (**E**) waveforms averaged by inter-pulse interval from representative experiments.

**Figure 2 toxics-09-00206-f002:**
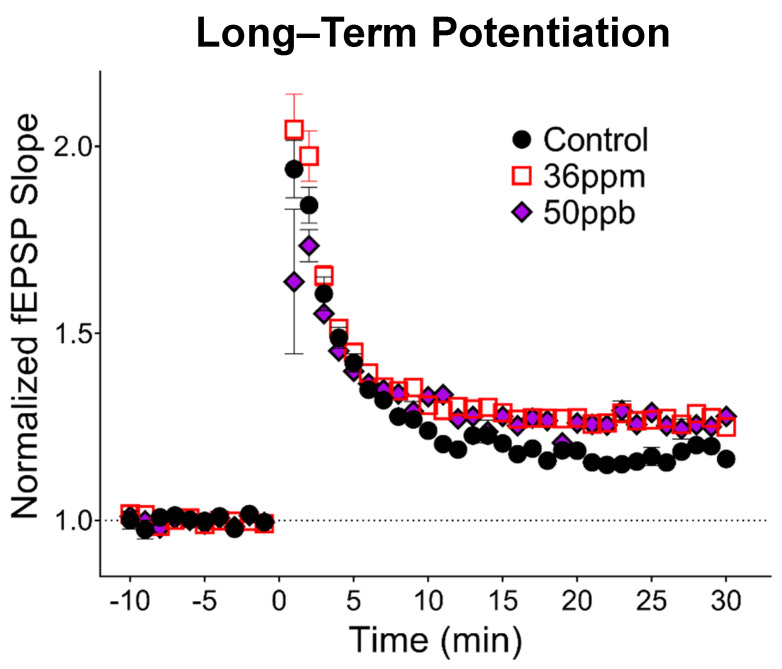
Effect of arsenic on hippocampal synaptic plasticity. Long-term potentiation induced by 1 × 100 Hz stimulation at time 0. Responses have been binned by one-minute intervals to aid visualization (average of four responses per minute).

**Table 1 toxics-09-00206-t001:** Timeline of arsenic exposure in current and previous electrophysiology studies.

Study	Exposure Type *	Age of Rodents Tested ^†^	Exposure Onset	Exposure End
Nelson-Mora et al. (2018)	In vivo	Adult	Gestation	Adulthood
Current Study	In vivo	Juvenile	Gestation	Early development
Kruger et al. (2006)	Ex vivo	Juvenile/Adult	Following brain slice preparation	NA
Kruger et al. (2007)	Ex vivo	Juvenile/Adult		
Kruger et al. (2009)	Ex vivo	Juvenile/Adult		

* In vivo exposure: rodents exposed to arsenic through drinking water; Ex vivo exposure: brain tissue exposed to arsenic metabolites after its removal from the rodent. ^†^ Juvenile mice: 14 to 23 days old; Adult mice: 2–4 months of age.

**Table 2 toxics-09-00206-t002:** Effect of arsenic exposure on synaptic function varies by exposure paradigm.

Study	Exposure Level	Basal Transmission	Paired-Pulse Facilitation (PPF)	Long-Term Potentiation (LTP)
In vivo exposure				
Nelson-Mora et al. (2018)	20 ppm	Decrease	Decrease *	Decrease
Current Study	36 ppm	Decrease	Decrease	Increase
Current study	50 ppb	Decrease	No change	No change ^†^
Acute, ex vivo exposure: Juvenile rodents				
Kruger et al. (2006)	1–100 μM	Decrease ^‡^	Not tested	No change
Kruger et al. (2007)	1–100 μM	Decrease	Not tested	Decrease ^‡^
Kruger et al. (2009)	1–100 μM	Decrease ^‡^	Not tested	Increase ^‡^
Acute, ex vivo exposure: Adult rodents				
Kruger et al. (2006)	0.1–100 μM	Decrease ^‡^	No change	Decrease
Kruger et al. (2007)	1–100 μM	Decrease	Not tested	Decrease ^‡^
Kruger et al. (2009)	10–100 μM	Decrease ^‡^	No change	Decrease

* Statistical test not performed. No change observed at short inter-pulse intervals (10–20 ms); possible decrease at other intervals. ^†^ Trend observed; warrants further study. ^‡^ Change not observed for all concentrations or metabolites tested.

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
