# Peer review of "Early Low-Level Arsenic Exposure Impacts Post-Synaptic Hippocampal Function in Juvenile Mice"

_toxics, 2021, doi:10.3390/toxics9090206_

Round 1
Reviewer 1 Report
Review of toxics Manuscript, ID: toxics-1297289
Early low-level arsenic exposure impacts post-synaptic hippocampal function in juvenile mice.
Foley, Barnett, Cory-Slechta & Xia
Brief Summary
This paper examined the effects of perinatal arsenic exposure on ex-vivo function of the hippocampus of juvenile mice. Two dose groups were used: 36ppm or 50ppb sodium arsenate delivered in the maternal drinking water. Exposure of the offspring was transplacental and via lactation. Hippocampal slices were collected from 17-23 day old offspring and electrophysiological indices of basal synaptic transmission and long-term potentiation were examined. Both arsenic levels decreased synaptic transmission, while increasing LTP.
Broad Comments
This is an interesting, well-written manuscript that aims to elucidate some of the neural mechanisms that underlie perinatal arsenic’s neurobehavioral toxicity. Arsenic is a significant, globally-distributed carcinogen and developmental toxicant that merits ongoing investigation. Strengths of the current manuscript include the authors’ use of environmentally relevant exposure levels and administration through contaminated drinking water. Their focus on hippocampal function is justified since perinatal arsenic is known to impair learning and attention in humans and laboratory animals and it is well-established that these behaviors require an intact hippocampus.
This paper should be of interest to readers of toxics after the authors respond to the comments and suggestions listed below. Most of the comments are minor and involve presentation style and word-choice concerns. Two comments are more substantial and should be addressed prior to publication: (1) revise and explain the role of sex in the statistical analysis (more detail below) (2) expand the Discussion section to contextualize and interpret the results. Currently, the Discussion section is merely a recapitulation of the Results section. What are the likely functional consequences of the hippocampal changes observed in the current study? What do the current effects in the juvenile brain mean in regards to the ongoing development of the brain? And, most importantly, how should the reader interpret an arsenic-induced increase in LTP? I am assuming that the authors are not suggesting that perinatal arsenic can function as cognitive enhancer?!
Specific Comments
Line 9 and 38: the authors use the phrase “all-cause mortality” in 2 places. Please replace this bit of jargon with something that is self-explanatory to a more general audience.
Lines 56-60: the authors review earlier findings that developmental arsenic decreases synaptic transmission and LTP. These findings contradict the increased LTP seen in the current study. The authors should address this in the Discussion section.
Lines 79-81: here, the authors describe a goal for the current study but the Introduction section should also contain an explicit hypothesis statement as well.
Table 1. I like the idea of the table but the current table is quite busy. Consider splitting table 1 into 2 tables. The first table should compare the exposure timelines of the cited studies (current, Nelson-Mora, Kruger). The second table should present just the columns containing the electrophysiological results.
Lines 100-102: Report the sample sizes. How many litters were prepared for the study? How many animals from each litter were used?
Lines 129-132: Explain why the males and females were pooled. If this decision was made to boost sample size in order to achieve statistical significance then indicate so in a straightforward manner. If not, the data should be analyzed with sex as a factor. Readers of toxics will be interested in the effects of sex regardless of whether or not it was a significant factor.
Figures 1 and 2. It is tough to distinguish the purple and red colors, especially in A2 and figure 2.
The authors administered sodium arsenate in the drinking water. There should be some mention of the arsenic cycle in the Discussion section. Is maternal methylation/inactivation likely to have happened?

Reviewer 2 Report
The current study demonstrated the effects of early developmental arsenic exposure in juvenile mice on synaptic transmission and plasticity in the hippocampus. The exposure to a low dose (50 ppb) of arsenic decreased basal synaptic transmission but no effect was observed in the pre-synaptic vesicular release. Overall, it is an interesting study that demonstrated the effects of the low dose of arsenic exposure. I have the following comments:
- I understood from figure legends that the number of biological replicates used for the study was: High exposure group (n =13), control (n = 7), and low-level exposure group (n = 7). Please include this information in the methods as well. Did the researchers calculate the power of the experiment? It looks like the sample size is small and why did authors have different numbers of replicates for each group?
- Why did authors pool males and females for all analyses? It would be interesting to see the effect of sex. Was the power of the experiment not enough to measure statistical significance if analyzed separately?
- It would be also interesting to see the effects of even a low dose of arsenic i.e., 10 ppb (since this is the current MCL) on basal synaptic transmission. Do authors have any plans to conduct these studies in future work?
- Page 3, Line 103: Please correct 23ºC
- Figure 2 legend: Please remove “arsenic on”. It was repeated twice.
Round 2
Reviewer 1 Report
The authors have addressed all of my concerns and comments.